# An HFman Probe-Based Multiplex Reverse Transcription Loop-Mediated Isothermal Amplification Assay for Simultaneous Detection of Hantaan and Seoul Viruses

**DOI:** 10.3390/diagnostics12081925

**Published:** 2022-08-10

**Authors:** Yi Zeng, Yun Feng, Yongjuan Zhao, Xiaoling Zhang, Lifen Yang, Juan Wang, Zihou Gao, Chiyu Zhang

**Affiliations:** 1Shanghai Public Health Clinical Center, Fudan University, Shanghai 201508, China; 2Yunnan Provincial Key Laboratory for Zoonosis Control and Prevention, Yunnan Institute of Endemic Diseases Control and Prevention, Dali 671000, China

**Keywords:** LAMP, Hantaan virus, Seoul virus, HFman probe, POCT

## Abstract

Hantaviruses are zoonotic pathogens that are widely distributed worldwide. Hantaan virus (HTNV) and Seoul virus (SEOV) are two most common hantaviruses that infect humans and cause hemorrhagic fever with renal syndrome (HFRS). Rapid and sensitive detection of HTNV and SEOV are crucial for surveillance, clinical treatment and management of HFRS. This study aimed to develop a rapid HFman probe-based mulstiplex reverse transcription loop-mediated isothermal amplification (RT-LAMP) assay to simultaneously detect HTNV and SEOV. A novel multiplex RT-LAMP assay was developed, and 46 serum samples obtained from clinically suspected patients were used for evaluation. The novel RT-LAMP assay can detect as low as 3 copies/reaction of hantaviruses with a detection limit of 41 and 73 copies per reaction for HTNV and SEOV, respectively. A clinical evaluation showed that the consistencies of the multiplex RT-LAMP with RT-qPCR assay were 100% and 97.8% for HTNV and SEOV, respectively. In view of the high prevalence of HTNV and SEOV in rural areas with high rodent density, a colorimetric visual determination method was also developed for point-of-care testing (POCT) for the diagnosis of the two viruses. The novel multiplex RT-LAMP assay is a sensitive, specific, and efficient method for simultaneously detecting HTNV and SEOV.

## 1. Introduction

Hantaviruses (family *Bunyaviridae*, genus *Orthohantavirus*) are zoonotic enveloped negative-stranded RNA viruses that can cause morbidity and mortality in humans [1,2,3]. The hantavirus genome includes three segments: L (large), M (medium), and S (small) segments that encode the viral RNA-depended RNA polymerase, envelope glycoproteins (Gn and Gc), and nucleoprotein (N), respectively [1,4]. The most serious human diseases caused by hantaviruses are hemorrhagic fever with renal syndrome (HFRS) and hantavirus cardiopulmonary syndrome (HCPS) [5,6,7]. Hantaviruses include some genotype/serotypes, including Hantaan virus (HTNV), Seoul virus (SEOV), Puumala virus (PUUV), Dobrava virus (DOBV), etc [8,9], and HTNV and SEOV are the major hantaviruses causing HFRS in Asia (especially in China) [10,11]. HTNV-associated HFRS is often more severe with a mortality rate of 5–10%, while SEOV-associated HFRS is characterized by mild symptoms [12].

China is the country with the most serious HFRS epidemic in the world with 40,000–60,000 cases reported annually [13]. The incidence rate of HFRS cases is estimated to be 0.5785 to 3.93 per 100,000 persons in China [14,15], and a total of 209,209 HFRS cases and 1855 deaths have been reported in the country, with the death rate of 0.89% [16]. Currently, no effective treatments are available for HFRS [17]. Furthermore, in spite of some progression in the development of an HFRS vaccine (especially the DNA vaccine candidates) and the approval of whole-virus inactivated vaccines against HTNV or SEOV in Korea and China, the protective efficacy of these vaccines and/or candidates need to be critically evaluated [18,19]. Therefore, HFRS has become a significant public health problem in China [11]. The rapid, accurate, simple, and reliable diagnosis of HTNV and SEOV infections plays a crucial role in the prevention and control of HFRS. 

Various diagnostic methods have been developed for detecting hantavirus infection, including serological methods [20,21,22,23] and nucleic acid detection methods [24,25,26]. Because of higher sensitivity and specificity than serological methods, several reverse transcription quantitative polymerase chain reaction (RT-qPCR) assays have been developed to detect hantaviruses [24,25,26] and can be used for early per-symptom diagnosis of HFRS. 

As zoonotic viruses, hantaviruses can be spread from their natural hosts to humans by aerosolized excreta inhalation [9]. *Apodemus agrarius* and *Rattus norvegicus* are the natural reservoirs of HTNV and SEOV [9]. Therefore, rural areas have a high prevalence of both HTNV and SEOV and contribute to more than 70% of HFRS cases [11,13]. However, the requirement of sophisticated instruments, and professional operators limits the implementation of RT-qPCR assays in rural areas, where rapid, simple, sensitive and accurate point-of-care (POC) diagnosis of HTNV and SEOV is most needed. Loop-mediated isothermal amplification (LAMP) method is a promising nucleic acid isothermal amplification method suitable for POC diagnosis of various pathogens [27,28,29]. In particular, our recent development of a multiplex, real-time, and variant-tolerant LAMP assay, which uses high-fidelity DNA polymerase to mediate florescent signal releasing from an HFman probe, not only largely improves the specificity and tolerance of LAMP assay to highly variable viruses, but also realizes the single pot multiplex detection of different targets [30]. In this study, we used the HFman probe-based RT-LAMP assay to rapidly and simultaneously detect HTNV and SEOV in a single tube. The novel assay is rapid, highly specific and sensitive for POC detection of HTNV and SEOV and can be used for routine sentinel surveillance in rural area.

## 2. Materials and Methods

### 2.1. Preparation of RNA Standard

PUC-57 plasmids containing the S gene of HTNV (GenBank: M14626.1) and SEOV (GenBank: AF488707.1) were synthesized by Sangon Biotech (Shanghai, China). To prepare the RNA standards, S genes were first amplified using specific PCR primers and then extracted with a DNA extraction kit (Monarch, NEB). RNA was in vitro synthesized via HiScribe T7 Quick High Yield RNA Synthesis Kit (New England Biolabs, English) and then DNase I was used to remove the DNA template. The obtained RNA was extracted and purified using alcohol and LiCl. The concentration of RNA standard was quantitated by Qubit 4.0 (Thermo Fisher Scientific) [31]. The copy number of each RNA was calculated using the following formula: RNA copies/mL = [RNA concentration (g/mL)/(nt transcript length × 340)] × 6.022 × 10^23^.

### 2.2. Primer Design

The LAMP primers and probes for HTNV and SEOV were designed through Primer Explorer V5 software (http://primerexplorer.jp/lampv5e/index.html, accessed on 10 June 2021) after multiple alignment using MEGA7 software. Primers were synthesized by Sangon Biotech (Shanghai, China). Each set of primers contained two external primers (F3 and B3), two internal primers (FIP and BIP), and loop primers (LB) (Appendix A). LB was synthesized as the HFman probe labeled with FAM/Cy5 fluorophore and a BHQ1/BHQ2 quencher group at 3′- and 5′-end of HTNV and SEOV, respectively (Table 1). 

### 2.3. Clinical Samples and RNA Extraction

A total of 46 serum samples collected by Yunnan Institute of Endemic Diseases Control & Prevention were included in this study to evaluate the performance of the novel RT-LAMP assay, with parallel RT-qPCR assay. The RNA was extracted using the TIANamp virus DNA/RNA kit (Tiangen Biotech, Beijing, China). In brief, RNA was extracted from 200 μL of serum, eluted in 50 μL nuclease-free H_2_O, and then stored at −80 °C until use. 

### 2.4. Single-Tube Multiplex RT-LAMP Reaction

The RT-LAMP was performed in 25 μL reaction that containing 1 × isothermal amplification buffer, 8 mM MgSO4, 1.8 mM dNTP, 8 U Bst 4.0 DNA/RNA polymerase (Haigene, China), 0.15 U High-fidelity DNA polymerase (New England Biolabs, Beverly, MA, USA), the primer mix for both HTNV and SEOV, including 0.1 μM each of F3 and B3, 1.0 μM each of FIP and BIP, 0.6 μM LB (or 0.3 μM LB and 0.3 μM probe), and 3 μL of template. The reaction was incubated at 64 °C for 50 min. Fluorescence signal was collected using the CFX 1000 Touch Real-Time PCR Detection System (Bio-Rad Laboratories, Hercules, CA, USA) each minute.

### 2.5. RT-qPCR Assay

To evaluate the performance of the multiplex RT-LAMP assay, a multiplex RT-qPCR assay was established for simultaneously detecting HTNV and SEOV according to a previously described multiplex RT-qPCR assay [24]. The primers and probes were modified from the previous paper and are shown in Table 1. The multiplex RT-qPCR reaction was performed using a One-Step Prime ScriptTM RT-PCR Kit (Takara, Dalian, China) in a CFX 1000 Touch Real-Time PCR Detection System (Bio-Rad Laboratories, USA). The 25 μL mixture contained 12.5 μL of 2× One-Step RT-PCR buffer, 0.5 μL of Ex Taq HS (5 U/μL), 0.5 μL of PrimeScript RT enzyme mix, 0.4 mΜ each of HASE-F and -R, 0.25 μM HTNV-P/SEOV-P, and 3 μL template. Reactions were incubated at 42 °C for 5 min and then 95 °C for 10 s, followed by 40 cycles of 95 °C for 5 s and 60 °C for 30 s. Samples were considered positive if they had a Ct value less than 40 [32]. 

### 2.6. Sensitivity and Specificity Test

The specificity of the multiplex RT-LAMP assay was assessed using 7 common human viruses, including influenza A; HCoV-229E; parainfluenza virus 3; human metapneumovirus; human rhinovirus; HCoV-HKU-1; and bocavirus. The sensitivity of the RT-LAMP assay was determined using ten-fold serial dilutions of RNA standard from 1.0 × 10^3^ to 1.0 × 10^0^ copies/μL.

### 2.7. Limit of Detection (LOD)

To determine the LOD of the multiplex RT-LMAP assay, 25 μL reactions with five-fold serial dilutions of RNA standard from 3000, 600, 120, 24, to 4.8 copies were performed. Each dilution was tested in 32 replicates. The LOD was defined as a 95% probability of obtaining a positive result using probit regression analysis with SPSS 17.0 software.

### 2.8. Colorimetric Assay

The colorimetric RT-LAMP was performed using the WarmStart^®^ Colorimetric LAMP 2× Master Mix (with cresol red) (New England Biolabs) and the LAMP primer set targeting the S gene of HTNV or SEOV (Table 1). The 25 μL mixture contained 12.5 μL of 2× Master Mix, 0.1 μM each of F3 and B3, 1.0 μM each of FIP and BIP, 0.6 μM LB, and 3 μL template. The reactions were performed at 64 °C, and the color change was observed at the 30, 40, and 50 min time points. The color change from burgundy to an orange or yellow color indicated positive results.

## 3. Results

### 3.1. Screening of Optimal RT-LAMP Primers

The S gene was the most conserved genomic region of HTNV and SEOV, and was subjected to the design of LAMP primers and probes using Primer Explorer. Four sets of primers were designed for detection of HTNV and SEOV using the RT-LAMP assay. To obtain the optimal primer set for the RT-LAMP assay, four sets of primers were subjected to a screening experiment under same conditions (reaction mix, primer concentration, reaction program, and same template input) as previously described [33] (Figure 1). Each primer set 1 was demonstrated to have highest amplification efficiency (with the lowest Tt values) (Figure 2 and Appendix A).

To further investigate the variation in the binding regions of the primer set 1 in corresponding virus, all available S gene sequences of HTNV and SEOV were downloaded from GenBank. After removing the low-quality sequences, 326 HTNV and 400 SEOV S gene sequences were subject to subsequent analyses. The results show that although the most conserved genomic region was used, a high variation was observed in the primer-binding regions (Appendix A). For example, there are some variations occurring in the 3′ ends of the primers F2, B2, F1c and B1c of HTNV. The presence of these variations might affect the detection sensitivity and rate of RT-LAMP assay, especially for some rare variants with mismatched based at the 3′-end of the primers. To improve the detection of various variants of HTNV and SEOV, we established an HFman probe-based multiplex RT-LAMP assay that can tolerate well the mismatches between primers and targets.

### 3.2. Sensitivity and Specificity of the Multiplex RT-LAMP Assay

A ten-fold serial diluted HTNV and SEOV RNA standards from 10^3^ to 1 copies/μL were used to determine the sensitivity of the multiplex RT-LAMP assay. The results show that the multiplex RT-LAMP can detect as few as 3 copies of HTNV and SEOV RNA per reaction within 15 min (Figure 3A). 

The specificity test showed that there was no amplification for other seven human viruses (including influenza A, HCoV-229E, parainfluenza virus 3, human metapneumovirus, human rhinovirus, HCoV-HKU-1, and bocavirus), indicating that the novel multiplex RT-LAMP assay is specific for HTNV and SEOV (Figure 3B).

For POC diagnosis in resource-limited settings, the RT-LAMP assay was further developed into a colorimetric format by adding cresol red. The interpretation of positive results has been described using this method. The sensitivity of the colorimetric assay for HTNV and SEOV was three copies per reaction (Figure 3C), similar to the real-time multiple RT-LAMP assay.

### 3.3. LOD of the Multiplex RT-LAMP Assay

We further measured the LOD of the multiplex RT-LAMP assay using serial dilution of RNA standards. The results show that all 32 reactions (100%) were positive for HTNV and SEOV RNA standards above 120 copies. When the template input was 24 and 5 copies, 24 and 11 of the 32 reaction replicates of HTNV, as well as 18 and 12 of the 32 reaction replicates of SEOV were positive, respectively. The LOD values of the assay were determined as 41 and 73 copies per reaction for HTNV and SEOV, respectively (Table 2). 

### 3.4. Clinical Evaluation

To validate the clinical application of the multiplex RT-LAMP assay, a comparison with a previously described RT-qPCR assay [24] was performed using 46 clinical samples collected from HFRS-suspected patients. Among these samples, only one was detected as SEOV positive by the RT-qPCR assay, whereas two were detected as SEOV-positive by the multiplex RT-LAMP assay (Table 3). Neither the RT-LAMP nor the RT-qPCR assays detected were HTNV-positive. The consistency of the two assays for HTNV and SEOV detection was 100%, and 97.8%, respectively. 

## 4. Discussion

Hantaviruses are important zoonotic pathogens with a growing global concern [2,3]. Rodents are the natural reservoirs of hantaviruses [9]. In China, HFRS caused by HTNV and SEOV is seriously threating local humans in rural areas with poor health care systems [11]. The disease caused by HTNV is relatively more severe and has a morbidity rate of 5–10%, higher than that caused by SEOV infection [12]. The development of a rapid, simple and highly sensitive assay for POC diagnosis of HFRS infection in the remote and rural areas is crucial for the prevention and control of HFRS. In this study, a rapid HFman probe-based multiplex RT-LAMP assay was developed to simultaneously detect HTNV and SEOV.

Because the LAMP method has a comparable sensitivity and a faster amplification speed than RT-qPCR [24,32]—its results can be easily observed by naked eye with formats of color changes [29], fluorescent signal [30], or magnesium pyrophosphate precipitation [34]—it is considered a promising tool for the diagnosis of various pathogens, including bacteria and viruses [27,31,35]. However, frequent non-specific amplification and lack of capacity of multiplex detection in single-tube limits the commercial use of LAMP and other isothermal methods [36,37]. We recently developed an HFman probe-based multiplex RT-LAMP method, which uses high-fidelity DNA polymerase to recognize and cleave an HFman probe to achieve highly sensitive, specific and multiplex detection in single-pot format [30]. Furthermore, the novel multiplex LAMP system is especially suitable for the rapid detection of highly variable viruses (e.g., HIV-1) [38,39]. Two RT-LAMP assays were previously reported to detect hantaviruses [32,40]. However, these assays had a relatively low sensitivity and were unable to simultaneously detect and distinguish HTNV and SEOV in a single tube.

The S gene of hantaviruses encodes nucleoproteins and a variation in the S gene might be associated with the antigenicity and virulence of hantaviruses [41]. Because of the relatively high homology among different serotypes [42], the S gene was frequently used as a primary target for hantavirus detection assays [24,32]. In this study, the S gene was also used as the primary target for the development of the multiplex RT-LAMP assay, and the optimal primer set was selected from four sets of newly designed primers. The sensitivity of the novel RT-LAMP assay was three copies of HTNV and SEOV per reaction, slightly superior to previous RT-LAMP and RT-qPCR assays [25,26,32,40].

The LOD of the novel multiplex RT-LAMP assay was 41 and 73 copies per reaction for HTNV and SEOV, respectively, and no cross-reactivity between HTNV and SEOV, as well as with seven human viruses was observed, indicating a high specificity. In particular, when the template input is more than the LOD, the assay has a very fast detection speed of less than 20 min. Therefore, we also developed a colorimetric format for the monitoring and/or surveillance of the prevalence of both viruses among rodents in the field. The sensitivity of the colorimetric RT-LAMP assay was also three copies per reaction, same to the multiplex real-time RT-LAMP assay. 

The clinical evaluation of the multiplex RT-LAMP assay was performed using 46 clinical samples collected from HFRS-suspected patients. The multiplex RT-LAMP assay showed a high consistency with RT-qPCR assays of 100% and 97.8% for HTNV and SEOV detection, respectively. In particular, the multiplex RT-LAMP assay detected one more positive for SEOV than the RT-qPCR assay, possibly implying a higher sensitivity. A major limitation of this study was the small sample size of the clinical samples, especially the positive samples for both viruses. A further evaluation using more samples might be needed in the future. 

## 5. Conclusions

The novel multiplex RT-LAMP method established in this study can rapidly and accurately detect and distinguish HTNV and SEOV in a single tube. The entire diagnostic procedure can be performed within 50 min and no sophisticated laboratory equipment is required. Furthermore, a colorimetric format can be used for POCT detection of both viruses in field and other resource-limited settings.

## Figures and Tables

**Figure 1 diagnostics-12-01925-f001:**
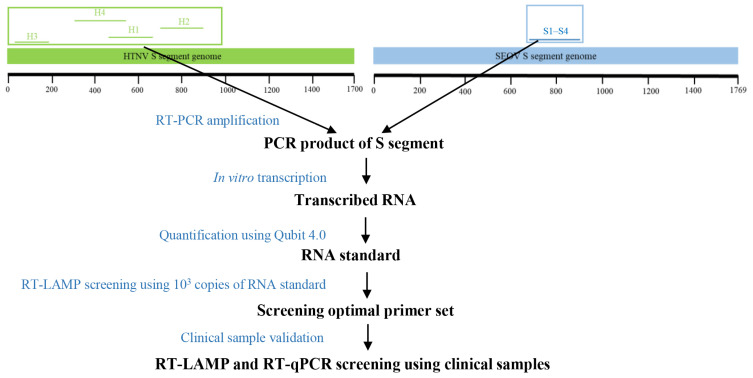
Genomic location and screening strategy of RT-LAMP primers for HTNV and SEOV. The location of each primer set is detailed in Appendix A.

**Figure 2 diagnostics-12-01925-f002:**
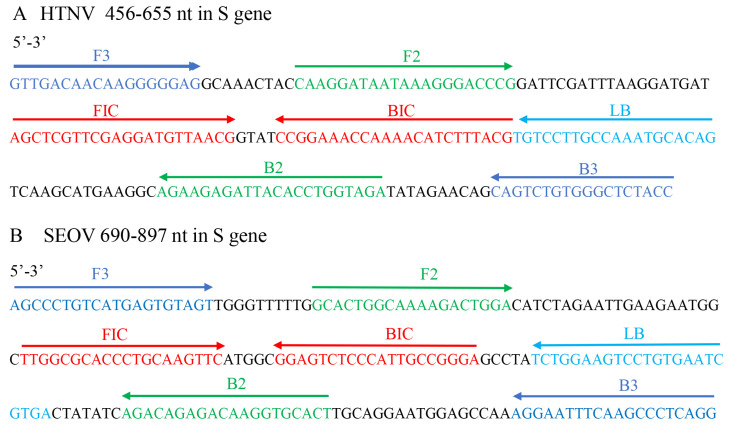
The genomic location of the RT-LAMP primers in the S gene of HTNV and SEOV. The primers include F3, B3 (outer primer), F2, F1c, B2, and B1c (inner primers), and LB (loop primer).

**Figure 3 diagnostics-12-01925-f003:**
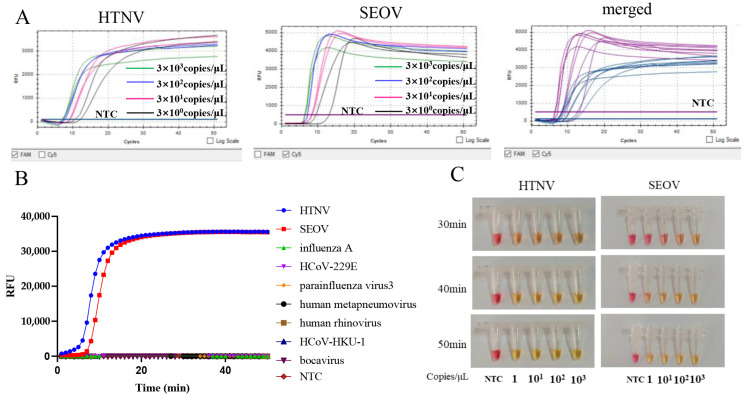
Sensitivity and specificity of the multiplex RT-LAMP assay. (**A**) Sensitivity. The RT-LAMP reaction was carried out with serial dilutions of 3 × 10^3^–3 × 10^0^ copies/μL of HTNV and SEOV standard RNA. (**B**) Specificity. Tested viruses included influenza A, HCoV-229E, parainfluenza virus 3, human metapneumovirus, human rhinovirus, HCoV-HKU-1, and bocavirus. (**C**) Colorimetric reaction of the RT-LAMP assay for HTNV and SEOV. The color change from burgundy to orange or yellow was considered as positive. NTC, non-template control.

**Table 1 diagnostics-12-01925-t001:** Primers and probes used in this study.

Methods	Primer Sets	Primer Sequence (5′-3′)	Refs.
RT-LAMP for -HTNV	H1-F3	GTTGACAACAAGGGGGAG	This study
	H1-B3	GGTAGAGCCCACAGACTG	
	H1-FIP	CGTTAACATCCTCGAAC-GAGCT-CAAGGATAATAAAGGGACCCG	
	H1-BIP	CCGGAAAC-CAAAACATCTTTACG-TCTACCAGGTGTAATCTCTTCT	
	H1-LB	TGTCCTTGCCAAATGCACAG	
	H1-HFman probe	BHQ1-TGTCCTTGCCAAATGCACAG-FAM	
RT-LAMP forSEOV	S1-F3	AGCCCTGTCATGAGTGTAGT	This study
	S1-B3	CCTGAGGGCTTGAAATTCCT	
	S1-FIP	GAACTT-GCAGGGTGCGCCAA-GCACTGGCAAAAGACTGGA	
	S1-BIP	GGAGTCTCCCATTGCCGG-GA-AGTGCACCTTGTCTCTGTCT	
	S1-LB	TCTGGAAGTCCTGTGAATCGTGA	
	S1-HFman probe	BHQ2-TCTGGAAGTCCTGTGAATCGTGA-Cy5	
RT-qPCR	HASE-F	GWGGVCARACAGCWGAYT	Modified from ref. [24]
	HASE-R	TCCWGGTGTAADYTCHTCWGC	
	SEOV-P	Cy5-CCATAATTGTCTATCTGACATCA-BHQ2	
	HTNV-P	FAM-AGCATCATCGTCTATCTTACATCC-BHQ1	

**Table 2 diagnostics-12-01925-t002:** Detection limit of RT-LAMP for HTNV and SEOV.

Template Input (Copies/25 μL Reaction)	HNTV (Positive/Total)	SEOV (Positive/Total)
3000	32/32	32/32
600	32/32	32/32
120	32/32	32/32
24	24/32	18/32
5	11/32	12/32
LOD (copies/25 μL reaction)	41	73

**Table 3 diagnostics-12-01925-t003:** Comparison of the multiplex RT-LAMP assay with RT-qPCR assay using HFRS-suspected patients.

Viruses	RT-LAMP (+/−)	RT-PCR (+/−)	Concordance Rate (%)
HTNV positive	0/46	0/46	100%
SEOV positive	2/44	1/45	97.8%
Total	46	46	

## Data Availability

The data presented in this study are available on request from the corresponding author.

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
