# Peer review of "An HFman Probe-Based Multiplex Reverse Transcription Loop-Mediated Isothermal Amplification Assay for Simultaneous Detection of Hantaan and Seoul Viruses"

_diagnostics, 2022, doi:10.3390/diagnostics12081925_

Round 1

Reviewer 1 Report

The development of simple and reliable methods for detecting orthohantavirus RNA is undoubtedly an important and urgent task. The method proposed by the authors of the manuscript may be in demand and find wide application for the rapid detection of HNTV and SEOV in suspected HFRS cases, especially in rural areas point-of-care diagnosis. One can only regret that the authors did not test the proposed method on virus-positive clinical samples, which makes it difficult to evaluate the effectiveness of the method in real point-of-care conditions.

Howewer, I have a few comments.

1. References 11 and 13 looks too outdated. It is necessary to provide the current information, for example, on the incidence of HFRS in China over the past 5-10 years.

2. The nucleotide sequences of H1-Hfman probe and S1-Hfman probe shown in Table 1 are completely identical. However, for the SEOV strain sequence (AF488707), the annealing site corresponding to the sequence S1-Hfman probe (578-597) is located outside the site amplified by Primer set S1 (691-897). Is this the correct sequence for S1-Hfman probe?

Minor comments:

1. In table 1, the rows in column 2 (Primer sets) do not quite match the rows in column 3 (Primer sequence), which makes it difficult to understand.

2. Line 134 – typo «3. Results ».

Author Response

We thank the reviewer for the valuable comments and suggestions. We have revised the manuscript according to your suggestions and marked the new changes in red. Please see the attachment.

The development of simple and reliable methods for detecting orthohantavirus RNA is undoubtedly an important and urgent task. The method proposed by the authors of the manuscript may be in demand and find wide application for the rapid detection of HNTV and SEOV in suspected HFRS cases, especially in rural areas point-of-care diagnosis. One can only regret that the authors did not test the proposed method on virus-positive clinical samples, which makes it difficult to evaluate the effectiveness of the method in real point-of-care conditions.

Howewer, I have a few comments.

Point 1: References 11 and 13 looks too outdated. It is necessary to provide the current information, for example, on the incidence of HFRS in China over the past 5-10 years.

Response 1: Thank you for your suggestion. We updated the current information of HFRS incidence in China (The incidence rate of HFRS cases was estimated to be 0.5785 to 3.93 per 100,000 persons in China [14,15], and a total of 209,209 HFRS cases and 1,855 deaths were reported in China, with the death rate of 0.89% [16]) (lines 44-46).

Point 2: The nucleotide sequences of H1-Hfman probe and S1-Hfman probe shown in Table 1 are completely identical. However, for the SEOV strain sequence (AF488707), the annealing site corresponding to the sequence S1-Hfman probe (578-597) is located outside the site amplified by Primer set S1 (691-897). Is this the correct sequence for S1-Hfman probe?

Response 2: Thank you for pointing out this error. We corrected the sequence of the S1-HFman probe in Table 1.

Minor comments:

Point 3: In table 1, the rows in column 2 (Primer sets) do not quite match the rows in column 3 (Primer sequence), which makes it difficult to understand.

Response 3: Thank you. We revised this table.

Point 4: Line 134 – typo «3. Results ».

Response 4: The "3. Results " has been deleted.

Reviewer 2 Report

Broad Comments:

This original manuscript takes an approach into diagnostic line to contribute to the successful diagnostic of orthohantaviruses. This is the valuable addition to the field of science and diagnostic medicine. In general, this manuscript is easy to read and follow. The figures and the text throw together. Rapid diagnostics are needed more especially in areas where there are few health centers and during the time of outbreak of infections. The manuscript will benefit from attending to few specific comments bellow.

Specific Comments:

Line 15: The authors should check the current literature on what viruses cause HFRS and HPS. Hantaan and Seoul orthohantaviruses are old world viruses and do not cause HPV but HFRS. The authors should consider rewriting the sentence.

Line 17: Since the authors are writing about HTNV and SEOV then the mentioning of HPS in the sentence makes hard reading.

Line 31: The authors check the correct systematic of viruses on the ICTV.

Throughout the manuscript the authors could consider checking the use of hantavirus as a genus system, according to the ICTV hantavirus genus was named to orthohantavirus

Lines 44-45: The authors could check Connie S Schmaljohn, 2014 on vaccines developed in China

Lines 89-90: It is not clear to which RT-qPCR the authors are referring to.

Lines 101-103: It is not clear whether the authors mixed both primers for HTNV and SEOV in one tube.

Lines 130-133: The authors should check carefully the sentences under these lines. There some repetition with what was stated in the Single-tube multiplex RT-LAMP reaction paragraph.

Lines 133-134: typo “should read: indicate positive results”.

Line 134: Also, typo “.3. Results”

Figure 2:  The figure will improve reading if the authors could add the position of these genes on the S segment genome

Line 152: it is not clear whether 326 HTNV and 400 SEOV S gene sequences refer to the length of S gene or the number of S gene downloaded. The author should consider paraphrasing this sentence to reflect the meaning.

Line 193: Typo a pre viously should ready “previously”

Table 4: It will improve reading of the table legend if “using HFRS-suspected patients” can be added.

Line 204: a high density of rodents makes it difficult to read, the authors should consider removing “a high density of rodents” since the preceding lines are talking about the disease HFRS.

Line 206: What does wild mean here?

Line 208: The authors should check this sentence “for simultaneously detect HTNV and SEOV” its either it reads “for simultaneously detecting HTNV and SEOV” or to “simultaneously detect HTNV and SEOV”.

Lines 210-213: These sentences make it difficult reading the authors should consider making short sentences from it and they should check the use of “but”. The sentence is not clear whether the authors are listing the specificities between LAMP method and RT-qPCR.

Line 246: Typo “a” it should be in capital latter “A”

Line 252: The acronym point-of-care (POC) is defined but I could not see the definition of POCT.

Author Response

 We thank the reviewer for the valuable comments and suggestions. We have considered all your comments and suggestions and revised the manuscript. We have marked the new changes in red. Please see the attachment.

Broad Comments:

This original manuscript takes an approach into diagnostic line to contribute to the successful diagnostic of orthohantaviruses. This is the valuable addition to the field of science and diagnostic medicine. In general, this manuscript is easy to read and follow. The figures and the text throw together. Rapid diagnostics are needed more especially in areas where there are few health centers and during the time of outbreak of infections. The manuscript will benefit from attending to few specific comments bellow.

Specific Comments:

Point 1: Line 15: The authors should check the current literature on what viruses cause HFRS and HPS. Hantaan and Seoul orthohantaviruses are old world viruses and do not cause HPV but HFRS. The authors should consider rewriting the sentence.

Response 1: Thank you for pointing out this error. Indeed, Hantaan and Seoul orthohantaviruses are not associated with HPS (Clinical microbiology and infection 2019, 21s, e6-e16). We removed HPS from all involved sentences throughout the paper.

Point 2: Line 17: Since the authors are writing about HTNV and SEOV then the mentioning of HPS in the sentence makes hard reading.

Response 2: HPS was removed from all involved sentences throughout the paper.

Point 3: Line 31: The authors check the correct systematic of viruses on the ICTV.

Throughout the manuscript the authors could consider checking the use of hantavirus as a genus system, according to the ICTV hantavirus genus was named to orthohantavirus

Response 3: Thank you for pointing out this error. We have corrected it throughout the paper (lines 31).

Point 4: Lines 44-45: The authors could check Connie S Schmaljohn, 2014 on vaccines developed in China

Response 4: Thank you for the suggestions on the information of HFRS vaccine developments. We cited the mentioned paper and revised the sentence as “Currently, no effective treatments are available for HFRS [17]. Furthermore, in spite of some progression in development of HFRS vaccine (especially the DNA vaccine candidates) and the approval of whole virus inactivated vaccines against HTNV or SEOV in Korea and China, the protective efficacy of these vaccines and/or candidates need to be critically evaluated [18-19]” (lines 46-51).

Point 5: Lines 89-90: It is not clear to which RT-qPCR the authors are referring to.

Response 5: The RT-qPCR assay was established based on a previous paper (Journal of virological methods 2005, 124, 21-26). The primers and probes descried in previous paper were used. We mentioned this in the Method section (lines 114 to 117) and Table 1.

Point 6: Lines 101-103: It is not clear whether the authors mixed both primers for HTNV and SEOV in one tube.

Response 6: Yes, we mixed the primer sets for HTNV and SEOV in one tube. For clarity, we revised the sentence as “...., the primer mix for both HTNV and SEOV, including 0.1μM each of F3 and B3, 1.0μM each of FIP and BIP, 0.6μM LB (or 0.3μM LB and 0.3μM probe), ...” (lines 108-109).

Point 7: Lines 130-133: The authors should check carefully the sentences under these lines. There some repetition with what was stated in the Single-tube multiplex RT-LAMP reaction paragraph.

Response 7: The single-tube multiplex RT-LAMP assay and the colorimetric assay are significantly different. The main differences included 1. different RT-LAMP reagents to be used, 2) mixed primer sets for both HTNV and SEOV to be used in the single-tube multiplex RT-LAMP reaction, while only one primer set for HTNV or SEOV to be used in a single tube colorimetric reaction, and 3) different loop primers and concentrations (0.3μM LB and 0.3μM HFman probe in the multiplex RT-LAMP assay, while only 0.6μM LB in the colorimetric assay) in both assays.

Point 8: Lines 133-134: typo “should read: indicate positive results”.

Response 8: Thank you. We corrected it (lines 143).

Point 9: Line 134: Also, typo “.3. Results”

Response 9: The "3. Results " has been deleted.

Point 10: Figure 2: The figure will improve reading if the authors could add the position of these genes on the S segment genome

Response 10: As suggested, we added the position of these genes on the S segment genome in Figure 2.

Point 11: Line 152: it is not clear whether 326 HTNV and 400 SEOV S gene sequences refer to the length of S gene or the number of S gene downloaded. The author should consider paraphrasing this sentence to reflect the meaning.

Response 11: As suggested, we revised the sentences as “To further investigate the variation of the binding regions of the primer set 1 in corresponding virus, all available S gene sequences of HTNV and SEOV were downloaded from GenBank. After removing the low-quality sequences, 326 HTNV and 400 SEOV S gene sequences were subject to subsequent analysis” (lines 154-157).

Point 12: Line 193: Typo a pre viously should ready “previously”

Response 12: It was corrected.

Point 13: Table 4: It will improve reading of the table legend if “using HFRS-suspected patients” can be added.

Response 13: As suggested, we added “using HFRS-suspected patients” in the title of Table 3.

Point 14: Line 204: a high density of rodents makes it difficult to read, the authors should consider removing “a high density of rodents” since the preceding lines are talking about the disease HFRS.

Response 14: It has been done as suggested.

Point 15: Line 206: What does wild mean here?

Response 15: We changed “wild” to “remote” in the revised manuscript (lines 217).

Point 16: Line 208: The authors should check this sentence “for simultaneously detect HTNV and SEOV” its either it reads “for simultaneously detecting HTNV and SEOV” or to “simultaneously detect HTNV and SEOV”.

Response 16: It has been corrected (lines 219).

Point 17: Lines 210-213: These sentences make it difficult reading the authors should consider making short sentences from it and they should check the use of “but”. The sentence is not clear whether the authors are listing the specificities between LAMP method and RT-qPCR.

Response 17: The “specificity” should be changed as “sensitivity”. We have revised this sentence as “Because the LAMP method has a comparable sensitivity and a faster amplification speed than RT-qPCR [24,32], and its results can be easily observed by naked eye with formats of color changes [29], fluorescent signal [30], or magnesium pyrophosphate precipitation [34], it is considered as a promising tool for diagnosis of various pathogens including bacteria and viruses [27,31,35]” (lines 221-225).

Point 18: Line 246: Typo “a” it should be in capital latter “A”

Response 18: It has been corrected.

Point 19: Line 252: The acronym point-of-care (POC) is defined but I could not see the definition of POCT.

Response 19: The definition of POCT (Point-of-care testing) is provided in Abstract.
